# Squalene Monooxygenase Gene *SsCI80130* Regulates *Sporisorium scitamineum* Mating/Filamentation and Pathogenicity

**DOI:** 10.3390/jof8050470

**Published:** 2022-04-30

**Authors:** Yichang Cai, Yi Zhang, Han Bao, Jiaoyun Chen, Jianwen Chen, Wankuan Shen

**Affiliations:** 1College of Agriculture, South China Agricultural University, Guangzhou 510642, China; caiyichang@stu.scau.edu.cn (Y.C.); zhangyiyi@stu.scau.edu.cn (Y.Z.); 543151973@stu.scau.edu.cn (H.B.); jiaoyunchen@stu.scau.edu.cn (J.C.); chenjianwen@scau.edu.cn (J.C.); 2Sugarcane Research Laboratory, South China Agricultural University, Guangzhou 510642, China; 3Scientific Observing and Experimental Station of Crop Cultivation in South China, Ministry of Agriculture and Rural Areas, Guangzhou 510642, China

**Keywords:** *Sporisorium scitamineum*, squalene monooxygenase, ergosterol, sexual mating, pathogenicity

## Abstract

Sugarcane is an important sugar crop and energy crop worldwide. Sugarcane smut caused by *Sporisorium scitamineum* is a serious fungal disease that occurs worldwide, seriously affecting the yield and quality of sugarcane. It is essential to reveal the molecular pathogenesis of *S. scitamineum* to explore a new control strategy of sugarcane smut. Based on transcriptome sequencing data of two *S. scitamineum* strains *Ss16* and *Ss47*, each with a different pathogenicity, our laboratory screened out the *SsCI80130* gene predicted to encode squalene monooxygenase. In this study, we obtained the knockout mutants *(ΔSs80130^+^* and *ΔSs80130*^−^) and complementary mutants (*COM80130*^+^ and *COM80130*^−^) of this gene by the polyethylene glycol-mediated (PEG-mediated) protoplast transformation technology, and then performed a functional analysis of the gene. The results showed that the deletion of the *SsCI80130* gene resulted in the increased content of squalene (substrate for squalene monooxygenase) and decreased content of ergosterol (the final product of the ergosterol synthesis pathway) in *S. scitamineum*. Meanwhile, the sporidial growth rate of the knockout mutants was significantly slower than that of the wild type and complementary mutants; under cell-wall stress or oxidative stress, the growth of the knockout mutants was significantly inhibited. In addition, the sexual mating ability and pathogenicity of knockout mutants were significantly weakened, while the sexual mating ability could be restored by adding exogenous small-molecular signal substance cAMP (cyclic adenosine monophosphate) or tryptophol. It is speculated that the *SsCI80130* gene was involved in the ergosterol biosynthesis in *S. scitamineum* and played an important role in the sporidial growth, stress response to different abiotic stresses (including cell wall stress and oxidative stress), sexual mating/filamentation and pathogenicity. Moreover, the *SsCI80130* gene may affect the sexual mating and pathogenicity of *S. scitamineum* by regulating the ergosterol synthesis and the synthesis of the small-molecular signal substance cAMP or tryptophol required for sexual mating. This study reveals for the first time that the gene encoding squalene monooxygenase is involved in regulating the sexual mating and pathogenicity of *S. scitamineum*, providing a basis for the molecular pathogenic mechanism of *S. scitamineum*.

## 1. Introduction

Sugarcane is the most important sugar and energy crop in the world and an important source of edible sugar, wildly growing in tropical and subtropical regions [1,2]. Sugarcane smut is a fungal disease caused by *Sporisorium scitamineum*, which seriously hinders the normal growth of sugarcane. The disease was first discovered in Natal, South Africa in 1877, and then gradually spread to sugarcane planting areas around the world [3]. In China, several pandemics of sugarcane smut have occurred, causing extremely serious economic losses, and the differentiation of *S. scitamineum* physiological races can lead to the disappearance of resistance in disease-resistant varieties [4]. Main sugarcane cultivars are facing the crisis of being eliminated, which seriously hinders the sustainable development of the Chinese sugarcane industry [5]. Sugarcane is easily infected by *S. scitamineum* at the early growth stage, and the infected plants show symptoms such as increased tillering, slender leaves, and small stems [6]. During the continuous infection of *S. scitamineum*, the top of the diseased cane plant gradually pulls out the whip-like material composed of plant tissue and chlamydospores, which is silver white in the early stage; after a few months, a large number of sporidia are produced and then form a black whip several tens of centimeters in length [7].

*Sporisorium scitamineum* is a typical dimorphic fungus and can be divided into two forms: haploid and dikaryotic mycelium [8]. Haploid sporidia are in the shape of long oval rods and have two opposite mating types: “+” and “−” [9]. A single type of sporidium is not pathogenic, and only when two sporidia of opposite mating type form dikaryotic hyphae through sexual mating can *S. scitamineum* infect sugarcane and cause sugarcane smut [7,10]. The pathogenicity of *S. scitamineum* is closely related to the sexual mating of sporidia [11]. Based on the PEG-mediated protoplast transformation technology, several genes have been found to regulate the sexual mating and pathogenicity of *S. scitamineum*. The *Ram1* gene encoding farnesyltransferase regulated the sexual mating and pathogenicity of *S. scitamineum* and played an important role in cell-wall stability [12]. After knocking out the gene *SsKpp2* encoding a mitogen-activated protein kinase, the sexual mating ability of *S. scitamineum* was significantly reduced [13]. Wang et al. [14] found that the *SsAgc1* gene encoding the serine/threonine protein kinase was required for the sexual mating of *S. scitamineum*. The *SsPRF1* gene encoding the pheromone response factor was also involved in the regulation of sexual mating and pathogenicity in *S. scitamineum* [15].

There is also a study showing that the deletion of the *SsCI72380* gene (encoding sterol 14 alpha demethylase in ergosterol synthesis pathway) caused a partial blockade of ergosterol biosynthesis in *S. scitamineum*, which further resulted in the reduced sexual mating ability and pathogenicity [16]. Squalene monooxygenase (SQE) belongs to the flavoprotein monooxygenase family and participates in various oxidation reactions [17,18]. It oxidizes squalene to epoxysqualene in fungal ergosterol biosynthesis [19,20]. Ergosterol biosynthesis is critical for fungal growth [21,22,23]. CamPagnac et al. [24] found that under the action of fenpropimorph (a chemical inhibitor for SQE), the SQE activity of *Arbuscular mycorrhizal* was severely inhibited, resulting in partial obstruction of the ergosterol biosynthesis, reduced ergosterol content, suppressed hyphal growth, and reduced pathogenicity. In addition, allylamine antifungal drugs affected the early stage of ergosterol biosynthesis by inhibiting SQE, ultimately impeding the membrane function and cell-wall synthesis of *Candida albicans* [25]. The *ERG1* gene (encoding SQE) knockout mutants of *C. albicans* have a significantly reduced ergosterol content, resulting in the inability of the hyphae to grow normally [26].

SQE plays an important role in fungal growth and pathogenicity, but the functions of the gene encoding SQE in *S. scitamineum* have not been reported so far. Therefore, it is necessary to study this gene in *S. scitamineum*. Based on the transcriptome sequencing data of two different pathogenic strains, *Ss16* and *Ss47*, isolated in our laboratory [27], we screened out the gene *SsCI80130* predicted to encode the squalene monooxygenase whose expression level was significantly up-regulated in the strong pathogenic strain *Ss16*. In this study, the knockout mutants and complementary mutants of the *SsCI80130* gene were obtained using PEG-mediated protoplast transformation technology. By phenotypic analysis and pathogenic identification of the knockout mutants and complementary mutants, the biological function of the *SsCI80130* gene in *S. scitamineum* was discussed, and the molecular pathogenesis of *S. scitamineum* was clarified based on the gene *SsCI80130*.

## 2. Materials and Methods

### 2.1. Characterization of the SsCI80130 Gene Sequence

Based on previous transcriptome sequencing data in the laboratory, the gene *SsCI80130* predicted to encode squalene monooxygenase was identified as a significantly (*p* ≤ 0.05) differentially expressed gene in two strains, *Ss16* (strong pathogenicity) and *Ss47* (weak pathogenicity), of *S. scitamineum* [27]. The protein encoded by the *SsCI80130* gene was analyzed using the Compute pI/MW tool to determine the isoelectric point (pI) and molecular weight (MW) (https://web.expasy.org/compute_pi, 24 April 2021). Blast comparison based on the amino acid sequences (DNA sequences into amino acid sequences: https://www.novopro.cn/tools/translate.html, 24 April 2021) was performed on the NCBI database (https://blast.ncbi.nlm.nih.gov/Blast.cgi, 24 April 2021) to obtain the conserved domains of the protein encoded by the *SsCI80130* gene. Phylogenetic analysis of the protein encoded by the *SsCI80130* gene was carried out in MEGA 7, and the phylogenetic tree was drawn using the neighbor-joining method [28,29]. The subcellular localization of the protein encoded by the *SsCI80130* gene was predicted using the WoLF PSORT online tool (https://www.genscript.com/wolf-psort.html, 24 April 2021).

### 2.2. Strains and Growth Conditions

The wild-type haploid strains *Ss16^+^* and *Ss16*^−^ of *S. scitamineum* were isolated and identified in our laboratory and stored at −80 °C [30]. The culture medium used in this study included YePSA solid medium (yeast extract 1%, peptone 2%, sucrose 2%, and agar 2%), YePS liquid medium (yeast extraction 1%, peptone 2%, sucrose 2%, pH 7.0), YePS soft medium (yeast extract 1%, peptone 2%, sugar 2%, agar 0.7%), YePSS medium (yeast extract 1%, peptone 2%, sugar 2%, D-sorbitol 18.17%, agar 2%) and MM medium (K_2_HPO4 0.205%, KH_2_PO4 0.145%, NH_4_NO3 0.05%, (NH_4_)_2_SO4 0.03%, FeSO_4_ 0.001%, CaCl_2_ 0.001%, Glucose 0.2%, Z-Buffer 50%, pH 7.0). For the growth assay, sporidia of the wild type (*Ss16^+^* and *Ss16^−^*), knockout mutants (*ΔSs80130^+^* and *ΔSs80130*^−^), and complementary mutants (*COM80130^+^* and *COM80130*^−^) were cultured in 100 mL YePS liquid medium at 28 °C, shaking at 200 rpm for 24 h; aliquots of cultured sporidia were then diluted with fresh YePS liquid medium and the cell density was adjusted to 10^5^ cells per mL; sporidia were then cultured for another 48 h under the same conditions; the OD_600_ was measured with a spectrophotometer (NanoDrop 2000C) every 6 h to monitor the growth of the wild-type haploid sporidia, knockout mutants, or complementary mutants. For the mating/filamentation assay, equal volumes of the wild-type haploid sporidia, knockout mutants, or complementary mutants of opposite mating type were mixed and spotted on YePSA medium in the absence or presence of 5 mM cAMP or 0.02 mM tryptophol and then incubated in the dark at 28 °C for 42 h before photographing. For stress-tolerance assessment, the sporidial culture at OD_600_ = 1.0 and its 10-fold serial dilutions were inoculated on YePSA medium in the absence or presence of stress inducers, including 100 µg/mL Congo red (CR), 50 µg/mL SDS, 4 mM H_2_O_2_, and 500 mM NaCl, and then cultured in the dark at 28 °C for 48 h before assessment and photographing.

### 2.3. Nucleic Acid Manipulation

Genomic DNA was extracted using a modified CTAB method [31]. The PCR amplification was performed using Phanta Max super-Fidelity DNA Polymerase (Vazyme, P505). Purification of DNA fragments was conducted using a FastPure Gel DNA Extraction Mini Kit (Vazyme, DC301). Total RNA was extracted by TRIZOL (Vazyme, R401), and HiScript III RT SuperMix (Vazyme, R323) was used for cDNA synthesis. A NanoDrop ND-1000 (Thermo Fischer Scientific, Wilmington, DE, USA) was used for measuring the concentration and purity of nucleic acid.

### 2.4. Construction of the SsCI80130 Gene Knockout Mutants and Complementary Mutants

The deletion of the *SsCI80130* gene was performed according to the PEG-mediated protoplast transformation method that was using the hygromycin resistance gene (*Hpt*) to replace the target gene [32]. Two flanking fragments of the *SsCI80130* gene was PCR-amplified using wild-type *S. scitamineum* genomic DNA (*Ss16^+^* and *Ss16*^−^); after flanking fragments were fused with the upstream and downstream fragments of the *Hpt* gene, respectively, two fusion fragments were transformed into protoplasts to obtain the *SsCI80130* gene knockout mutants. The primer design of the amplified products was derived from the genome sequence LK056664.1 in NCBI.

The complementation of the *SsCI80130* gene was performed by a similar method [33]. The complemented gene not only carried the target gene to replace the *Hpt* gene in the knockout mutants but also carried the zeocin-resistance gene to screen. All primers involved in the construction and validation of the knockout mutants and complementary mutants are listed in Table 1.

### 2.5. Squalene Monooxygenase (SQE) Activity Assay

We determined the SQE activity every 12 h over a period of 48 h during the haploid sporidial growth and sexual mating culture (YePS, 28 °C, 200 rpm) as well as the sugarcane bud infection process (after inoculation of the smut susceptible sugarcane variety “ROC22”) to assess the effect of the *SsCI80130* gene on the SQE activity. The determination was carried out in accordance with the instruction of the SQE ELISA kit (Shanghai mlbio Enzyme-linked Biotechnology). Each group of experiments was repeated three times.

### 2.6. Determination of Squalene Content and Ergosterol Content

The determination method referred to Zhang et al. [34], with slight modifications. A total of 0.5 g sporidia colonies of the wild type (*Ss16^+^* and *Ss16*^−^), knockout mutants (*ΔSs80130*^+^ and *ΔSs80130*^−^), or complementary mutant (*COM80130*^+^ and *COM80130*^−^) collected from YePS liquid medium (cultured at 28 °C for 48 h) were saponified by 10 mL saponification solution (50% KOH solution:absolute ethanol = 2:3) in a water bath of 88 °C for three hours. The products were then extracted with petroleum ether. After evaporation drying, samples were dissolved in N-hexane. Using ergosterol as an external standard, gas chromatography (GC) was used to determine the squalene content and ergosterol content (mg/g). The chromatographic determination conditions were as follows: Shimadzu gas chromatograph (Agilent 7890B); capillary column DB-5 (25.0 m × 0.25 mm × 0.25 μm); temperature program: 195 °C, 3 min; rise to 300 °C with 5.5 °C per min; injector temperature: 280 °C; sample volume: 1 μL. Each group of experiments was repeated three times, and each sample was injected twice.

### 2.7. Analysis of Gene Expression

Quantitative real-time (qRT)-PCR was performed to detect the expression level of the key gene *Aro8* in tryptophol synthesis and the key gene *Uac1* in cAMP synthesis during the sexual mating culture [35]. Three sexual mating combinations, including the wild-type combination (*Ss16*^+^ + *Ss16*^−^), knockout mutant combination (*∆Ss80130*^+^ + *∆Ss80130*^−^), and complementary mutant combination (*COM80130*^+^ + *COM80130*^−^), were designed. During 48 h of sexual mating cultivation, samples were taken every 12 h to extract the total RNA, and gene expression was detected by qRT-PCR. For the qRT-PCR, we used a ChamQ Universal SYBR qPCR Master Mix (Vazyme, Q711), and the reaction was run on a real-time PCR system (CFX96, BioRad). Relative expression values were calculated with the 2^−ΔΔCt^ method using *ACTIN* gene as an internal control [36]. Three biological repeats each containing three technical replicates for each sample were performed. In the same way, we assessed the expression level of the *SsCI80130* gene every 12 h over a period of 72 h during the haploid sporidial growth and sexual mating culture (YePS, 28 °C, 200 rpm) as well as sugarcane bud infection process (after inoculation of the smut-susceptible sugarcane variety “ROC22”) using qRT-PCR. Three biological repeats each containing three technical replicates for each sample were performed. All primers involved in qRT-PCR are listed in Table 1.

### 2.8. Assay of the Pathogenicity of the SsCI80130 Gene Knockout Mutants and Complementary Mutants

Sporidial colonies of the wild type (*Ss16*^+^ and *Ss16*^−^), knockout mutants (*ΔSs80130*^+^ and *ΔSs80130*^−^), or complementary mutants (*COM80130*^+^ and *COM80130*^−^) were inoculated into YePS liquid medium and cultured at 28 °C with shaking at 200 rpm for 24 h; sporidia were collected by centrifugation and washed twice with ddH_2_O, after which they were re-suspended in YePS liquid medium at a final concentration of 2 × 10^9^ sporidia/mL; sporidia of opposite mating type were mixed in an equal volume and then a 200 µL mixture was injected into the growth point of highly susceptible sugarcane variety “ROC22” (4–5 young leaves stage) with 15 plants in each treatment (5 plants per pot, 3 replicates); the wild-type combination (*Ss16*^+^ and *Ss16*^−^) served as a positive control, and sterile water was used as a negative control. Inoculated plants were placed in the greenhouse for a total of four months, and the incidence of sugarcane smut was observed and recorded one month after inoculation; diseased plants were marked during each investigation to avoid repeated investigations, and black whips were covered with a plastic bag to prevent the spread of sporidia. Finally, the number and incidence of diseased plants were counted.

### 2.9. Microscopy

Images were taken using an Axio Observer Z1 microscope (Zeiss, Jena, Germany) equipped with a sCMOS camera (PCO Edge, Kelheim, Germany).

### 2.10. Statistic Analysis

Data were expressed as the means ± standard error (SE). Differences among different treatments were analyzed using IBM SPSS Statistics 20.

## 3. Results

### 3.1. Identification and Characterization of the SsCI80130 Gene

The *SsCI80130* gene was identified as a significantly (*p* ≤ 0.05) differentially expressed gene in two *S. scitamineum* strains: *Ss16* (strong pathogenicity) and *Ss47* (weak pathogenicity) [27]. The *SsCI80130* gene was 1740 bp in length without introns; through NCBI annotation, the protein (NCBI: protein accession no. CDU24191.1) encoded by the *SsCI80130* gene was squalene monooxygenase and consisted of 579 amino acid residues; the isoelectric point (pI) of the protein was 8.85, and the molecular weight (MW) was 62.8 kDa (Figure 1A). Phylogenetic analysis showed that the protein encoded by the *SsCI80130* gene was highly homologous to *Sporisorium reilianum f.* sp. *reilianum* related to squalene monooxygenase and *Sporisorium reilianum SRZ2* related to squalene monooxygenase, indicating that the *SsCI80130* gene is highly conserved in smut fungi (Figure 1B). The subcellular localization of this protein was predicted to be on the endoplasmic reticulum (Figure 1C).

### 3.2. Molecular Construction of Knockout Mutants and Complementary Mutants

The construction of the *SsCI80130* gene knockout mutants and complementary mutants were performed as described in the Materials and Methods section. Using the wild-type (*Ss16^+^* and *Ss16*^−^) genomic DNA as a template, each flanking fragment was amplified by PCR with the primer pairs *SsCI80130*-LB-F/R and *SsCI80130*-RB-F/R, and the band sizes were 912 bp and 952 bp, respectively; meanwhile, the upstream fragment and downstream fragment of the *Hpt* gene were amplified by PCR with the primer pairs Hpt-LB-F/R and Hpt-RB-F/R, and the band sizes were about 2 Kb and 1.5 Kb, respectively (Figure 2A). Two flanking fragments were then fused with the upstream and downstream fragments of the *Hpt* gene, respectively, using the primer pairs *SsCI80130*-LB-F/Hpt-LB-R and Hpt-RB-F/*SsCI80130*-RB-R; the band sizes were about 3 Kb and 2.5 Kb, respectively (Figure 2B). Finally, the fusion fragments were transformed into the wild-type protoplasts. As expected, two knockout mutants (*ΔSs80130^+^* and *ΔSs80130*^−^) were obtained. The validation of knockout mutants was performed by PCR using the internal primer pair *SsCI80130*-IN-F/R and external primer pair *SsCI80130*-OU-F/R; the internal primer pair produced a 1084 bp band from the *SsCI80130* gene in the wild type (*Ss16^+^* and *Ss16*^−^), but no band in the knockout mutants; the external primer pair produced a 5011 bp band in the wild type but a 6419 bp band in knockout mutants due to the insertion of the *Hpt* gene (Figure 2C,D). Similarly, in the PCR validation of the *SsCI80130* gene complementary mutants (*COM80130^+^* and *COM80130*^−^), the primer pair Zeocin-IN-F/R for zeocin resistance gene detection and the internal primer pair *SsCI80130*-IN-F/R produced a 503 bp band and a 1084 bp band, respectively, in the complementary mutants but no band in the knockout mutants (Figure 2E).

### 3.3. Effects of the SsCI80130 Gene on Morphology and Growth of S. scitamineum

It was observed that the sporidial colony morphology of the *SsCI80130* gene knockout mutants (*ΔSs80130^+^* and *ΔSs80130*^−^) on YePSA plates was not different from that of the wild-type (*Ss16^+^* and *Ss16*^−^) and complementary mutants (*COM80130^+^* and *COM80130*^−^) (Figure 3A). In contrast, the growth rate of the knockout mutants was found to be significantly slower than that of the wild type, while the growth rate of the complementary mutants basically returned to that of the wild type (Figure 3B,C). Under a microscope, the haploid sporidia of the knockout mutants, complementary mutant, and the wild type were all in the shape of one long oval rod (Figure 3D), which indicated that the deletion of the *SsCI80130* gene did not affect the *S. scitamineum* sporidial morphology.

### 3.4. Effects of the SsCI80130 Gene on Sexual Mating of S. scitamineum

On YePSA plates, the wild-type combination (*Ss16^+^* + *Ss16*^−^) and the combination containing complementary mutants (*COM80130^+^* + *COM80130*^−^, *Ss16^+^* + *COM80130*^−^ or *Ss16*^−^ + *COM80130^+^*) produced abundant white villous hyphae through sexual mating, while the sexual mating ability of the combination containing knockout mutants (*ΔSs80130^+^*+ *ΔSs80130*^−^, *Ss16^+^* + *ΔSs80130*^−^ or *Ss16*^−^ + *ΔSs80130^+^*) was basically lost, resulting in the inability to produce hyphae (Figure 4A). After the addition of small-molecular signaling substances (tryptophol or cAMP) that have been confirmed to be involved in the regulation of sexual mating in *S. scitamineum* [14,33], the sexual mating ability of the knockout mutants was fully restored, leading to the regeneration of white villous hyphae (Figure 4A). Furthermore, during the sexual mating culture, the expression level of the key gene *Aro8* related to tryptophol synthesis or the key gene *Uac1* related to cAMP synthesis in the knockout mutant combination (*ΔSs80130^+^* + *ΔSs80130*^−^) was significantly lower than that in the wild-type combination (*Ss16^+^* + *Ss16*^−^), while the gene-expression level in the complementary mutant combination (*COM80130^+^* + *COM80130*^−^) returned to the level of the wild-type combination (Figure 4B,C). The results indicated that *SsCI80130* gene might be involved in the synthesis or transport of tryptophol or cAMP. In addition, under a microscope, the hyphae produced by the combination containing knockout mutants after recovery of the sexual mating ability were not different from those produced by the combination containing the wild type or the complementary mutants under normal condition (Appendix A), indicating that the addition of tryptophol or cAMP does not affect the hyphal morphology of *S. scitamineum*.

### 3.5. Effects of the SsCI80130 Gene on the SQE Activity of S. scitamineum

During the growth of the haploid sporidia, the SQE activity of the knockout mutants (*ΔSs80130^+^* and *ΔSs80130*^−^) was always significantly lower than that of the wild type (*Ss16^+^* and *Ss16*^−^) and the complementary mutants (*COM80130^+^* and *COM80130*^−^) (Figure 5A). During the sexual mating culture, the SQE activity of the combination containing knockout mutants (*ΔSs80130^+^* + *ΔSs80130*^−^, *Ss16^+^* + *ΔSs80130*^−^ or *Ss16*^−^ + *ΔSs80130^+^*) was significantly lower than that of the wild-type combination (*Ss16^+^* + *Ss16*^−^) and the combination containing complementary mutants (*COM80130^+^* + *COM80130*^−^, *Ss16^+^* + *COM80130*^−^ or *Ss16*^−^ + *COM80130^+^*), while the SQE activity of the knockout mutant combination (*ΔSs80130^+^* + *ΔSs80130*^−^) was always the lowest (Figure 5B). During the sugarcane bud infection, the SQE activity was similar to that in the process of sexual mating culture (Figure 5C).

### 3.6. SsCI80130 Gene Is Involved in Ergosterol Biosynthesis of S. scitamineum

As the substrate for squalene monooxygenase, the squalene content in the knockout mutants (*ΔSs80130^+^* and *ΔSs80130*^−^) was significantly higher than that in the wild type (*Ss16^+^* and *Ss16*^−^), while the squalene content in the complementary mutants (*COM80130^+^* and *COM80130*^−^) returned to the wild-type level; on the contrary, the content of ergosterol (the final product of the ergosterol pathway) in the knockout mutants was significantly lower than that in the wild type and complementary mutants, indicating that the *SsCI80130* gene is involved in the ergosterol biosynthesis in *S. scitamineum* (Figure 6A–C). At the same time, it also confirmed that the protein encoded by the *SsCI80130* gene is squalene monooxygenase.

### 3.7. Effects of the SsCI80130 Gene on the Stress Tolerance of S. scitamineum

We examined the tolerance of the wild type, knockout mutants, and complementary mutants to various abiotic stresses by adding various substances, including cell-wall stress (SDS or Congo red), osmotic stress (NaCl), and oxidative stress (H_2_O_2_). The growth rates of the knockout mutants (*ΔSs80130^+^* and *ΔSs80130*^−^) on YePSA plates or MM plates were significantly slower than that of the wild type (*Ss16^+^* and *Ss16^−^*) and complementary mutants (*COM80130^+^* and *COM80130*^−^). After the addition of CR, SDS, or H_2_O_2_ to YePSA plates and MM plates, the growth rates of the knockout mutants were significantly inhibited compared to the wild type and complementary mutants, which indicated that the *SsCI80130* gene was involved in the stress response of *S. scitamineum* under cell-wall stress and oxidative stress (Figure 7).

### 3.8. SsCI80130 Gene Expression Level

During the culture of haploid sporidia, the expression level of the *SsCI80130* gene in the wild type (*Ss16^+^* and *Ss16*^−^) and complementary mutants (*COM80130^+^* and *COM80130*^−^) increased over time, peaking at 60 h and then decreasing slightly, while the gene expression was undetectable in the knockout mutants (*ΔSs80130^+^* and *ΔSs80130*^−^) (Figure 8A). During the sexual mating culture, the expression of the *SsCI80130* gene in the combination between the wild type and knockout mutant (*Ss16^+^* + *ΔSs80130*^−^ or *Ss16*^−^ + *ΔSs80130^+^*) was significantly lower than that in the wild-type combination (*Ss16^+^* + *Ss16*^−^) and the combination containing complementary mutants (*COM80130^+^* + *COM80130*^−^, *Ss16^+^* + *COM80130*^−^ or *Ss16*^−^ + *COM80130^+^*), while the gene expression was also not detected in the knockout mutants combination (*ΔSs80130^+^* + *ΔSs80130*^−^) (Figure 8B). During the sugarcane bud infection, the expression level of the *SsCI80130* gene was similar to that in the process of sexual mating culture (Figure 8C).

### 3.9. SsCI80130 Gene Is Required for the Pathogenicity of S. scitamineum

To identify whether the *SsCI80130* gene regulates *S. scitamineum* pathogenicity, the highly susceptible sugarcane variety “ROC22” was inoculated with sporidial mixtures of different combinations (opposite mating type) as follows: (*Ss16^+^ + Ss16*^−^), (*ΔSs80130^+^* + *ΔSs80130*^−^), (*Ss16^+^* + *ΔSs80130*^−^), (*Ss16*^−^ + *ΔSs80130^+^*), (*COM80130^+^ + COM80130*^−^), (*Ss16^+^ + COM80130*^−^) and (*Ss16*^−^ *+ COM80130^+^*); the wild-type combination (*Ss16^+^ + Ss16*^−^) was used as a positive control, and sterile water was used as a negative control (Figure 9A). After inoculation with sporidial mixtures of the combinations without the knockout mutants (*Ss16^+^ + Ss16*^−^, *COM80130^+^ + COM80130*^−^, *Ss16^+^ + COM80130*^−^ or *Ss16*^−^ *+ COM80130^+^*), cane plants showed high incidence, with about 83%, 83%, 75%, and 92%, respectively; in contrast, after inoculation with sporidial mixtures of the combinations containing knockout mutants (*ΔSs80130^+^* + *ΔSs80130*^−^, *Ss16^+^* + *ΔSs80130*^−^, or *Ss16*^−^ + *ΔSs80130^+^*), cane plants displayed significantly lower incidence, with only 0%, 8%, and 25%, respectively (Figure 9B). The result showed that the *SsCI80130* gene regulates *S. scitamineum* pathogenicity.

## 4. Discussion

In this study, based on the transcriptome sequencing data of two different pathogenic *S. scitamineum* strains, *Ss16* and *Ss47*, a *SsCI80130* gene predicted to encode squalene monooxygenase (SQE) was screened from the most enriched pathways in the Kyoto Encyclopedia of Genes and Genomes (KEGG) database. The knockout mutants and complementary mutants were then obtained by PEG-mediated protoplast transformation technology. During the construction of the mutants, we used the PCR verification method and did not perform Southern blot validation, so there might be some additional genomic changes. The phylogenetic tree showed that the *SsCI80130* gene was highly conserved among smut fungi. In addition, the subcellular localization of this protein was predicted to be on the endoplasmic reticulum (some tools predicted it to be located on the mitochondrial membrane, which needs further study. Li et al. [16] also found that different tools had different prediction results). In this study, the expression of the *SsCI80130* gene in the knockout mutants was not detected during the haploid sporidia culture, sexual mating culture, or sugarcane bud infection, while the expression level of the gene in the complementary mutants returned to that of the wild type. These results showed that the quality of knockout mutants and complementary mutants constructed in this study was reliable (the *SsCI80130* gene was completely deleted in the knockout mutants and almost completely restored in the complementary mutants), which was similar to a previous study on *S. scitamineum* by Li et al. [16,37]. The obtained high-quality knockout mutants and complementary mutants laid the foundation for further gene function research. Furthermore, during haploid sporidia culture, sexual mating culture, or sugarcane bud infection, the SQE activity of the knockout mutants were always significantly lower than that of the wild type and complementary mutants; the content of squalene (the substrate of the squalene monooxygenase predicted to be encoded by the *SsCI80130* gene) in the knockout mutant was significantly higher than that in the wild type and complementary mutants, while the ergosterol (the final product of the ergosterol pathway) content showed the opposite phenomenon, verifying that the protein encoded by the *SsCI80130* gene is squalene monooxygenase. Li et al. [16] also used the same method to verify that the protein encoded by the *SsCI72380* gene of *S. scitamineum* was the predicted protein.

Squalene monooxygenase (SQE) is a key enzyme in the fungal ergosterol biosynthesis pathway [38,39]. Ergosterol is a sterol-like compound commonly found in pathogenic fungi and plays an important role in cell membrane fluidity, cell growth, and material transport [40,41,42]. In S*accharomyces cerevisiae*, studies of the ergosterol biosynthesis pathway have mainly relied on the inhibition of the key enzymes’ activity by drugs or by knocking out genes encoding key enzymes, and so far, at least 25 enzymes have been found to be involved in ergosterol biosynthesis [43]. Ergosterol biosynthesis is closely related to the fungal growth and pathogenicity. Under the action of terbinafine, a specific inhibitor for SQE, the SQE activity of *C. albicans* was weakened and the ergosterol content was significantly reduced, resulting in significantly inhibited hyphal growth and weakened pathogenicity [25]. The *ERG1* gene (encoding SQE) knockout mutant in *C. albicans* had a significantly increased squalene content, decreased ergosterol content, and enhanced drug sensitivity, leading to inhibited hyphal growth [26]. Under the action of azoles, an antifungal drug targeting the key enzyme *Erg11A* in the ergosterol biosynthesis pathway, the ergosterol synthesis pathway in *Aspergillus fumigatus* was blocked, resulting in slower growth and weakened pathogenicity of the fungus [44]. In addition, Jin et al. [45] found that ergosterol regulates signal transduction and plasma membrane fusion during sexual mating in yeast. Furthermore, the blockade of ergosterol biosynthesis leads to a weakened sexual mating ability and reduced pathogenicity of *S. scitamineum* [16]. In this study, the *SsCI80130* gene knockout mutants of *S. scitamineum* had a significant decrease in SQE activity, a significant increase in squalene content, and a significant decrease in ergosterol content as well as a significant slowdown in the sporidial growth rate compared to the wild type and complementary mutants. Under the microscope, the sporidial morphology of the knockout mutants was not significantly different from that of the wild type and complementary mutants, which were both in the shape of a long oval rod. Moreover, the sexual mating ability of the knockout mutants was almost completely lost, and the pathogenicity was significantly reduced. These results indicated that the *SsCI80130* gene may regulate the growth rate, sexual mating, and pathogenicity of *S. scitamineum* by affecting ergosterol biosynthesis, but does not affect the sporidial morphology. This is basically consistent with the findings of Li et al. [16] that the ergosterol biosynthesis in *S. scitamineum* was involved in sexual mating and pathogenicity. In addition, as a typical dimorphic fungus, *S. scitamineum* is infective and pathogenic only after two haploid sporidia of opposite mating type sexually mate to produce dikaryotic hyphae, leading to the occurrence of sugarcane smut [4]. The cAMP signaling pathway is highly correlated with morphological transition and pathogenicity in dimorphic fungi [46,47]. The obstruction of cAMP synthesis affects the sexual mating of *S. scitamineum*, resulting in its inability to produce dikaryotic hyphae, but the sexual mating ability could be restored by the addition of exogenous small-molecular substance cAMP [13,33,37]. The cAMP signaling pathway also affects the morphological transition of *Ustilago maydis*. The deletion of the *Ubc1* gene (encoding the cAMP type II regulatory subunit) prevented *U. maydis* from producing hyphae [48]. Similarly, *C. albicans* could not switch from yeast form to hyphal form after the blockade of the cAMP synthesis pathway, resulting in a decrease in pathogenicity [49]. In addition, tryptophol, another small molecular substance as a quorum sensing molecule of a variety of fungi, also plays an important role in the sexual mating. Exogenous addition of tryptophol could restore the sexual mating ability of the *SsAgc1* (encoding AGC kinase) gene knockout mutant of *S. scitamineum* [14,50]. In order to further clarify how the *SsCI80130* gene regulates the pathogenic mechanism of *S. scitamineum*, we performed experiments with the exogenous addition of cAMP or tryptophol to knockout mutants. The results showed that the exogenous addition of 5 mM cAMP or 0.02 mM tryptophol could fully restore the sexual mating ability of the knockout mutants, resulting in the regeneration of white villiated hyphae. At the same time, the results of qRT-PCR showed that the expression level of the key gene *UAC1* related to cAMP synthesis or the key gene *Aro8* related to tryptophol synthesis in the knockout mutants was significantly lower than that of the wild type or complementary mutants. Furthermore, there was no obvious difference in the morphology of the recovered hyphae and the hyphae produced under the normal condition. The results showed that the *SsCI80130* gene may regulate the sexual mating and pathogenicity of *S. scitamineum* by regulating the cAMP signaling pathway and synthesis of the tryptophol, which is similar to the findings of Li et al. [37]. Moreover, in the abiotic-stress experiment with the addition of CR, SDS, or H_2_O_2_, the growth rate of the knockout mutants was significantly inhibited compared to the wild type and complementary mutants, indicating that the *SsCI80130* gene also plays an important role in maintaining the cell-wall integrity and oxidative-stress tolerance of *S. scitamineum*.

In conclusion, the *SsCI80130* gene knockout mutants and complementary mutants of *S. scitamineum* were obtained successfully by PEG-mediated protoplast transformation technology in this study. Comparative analysis of the growth rate, sporidial morphology, SQE activity, squalene content, ergosterol content, *SsCI80130* gene expression level, abiotic-stress tolerance, sexual mating ability, and pathogenicity showed that the *SsCI80130* gene (encoding squalene monooxygenase) in *S. scitamineum* is involved in ergosterol biosynthesis and regulates the sporidial growth and pathogenicity as well as playing an important role in oxidative-stress tolerance and maintaining the cell-wall integrity. Moreover, the *SsCI80130* gene may also regulate the sexual mating and pathogenicity of *S. scitamineum* by regulating the synthesis of small-molecular signal substances cAMP and tryptophol. This study reveals for the first time that the gene encoding squalene monooxygenase in the ergosterol synthesis pathway is involved in regulating the sexual mating and pathogenicity of *S. scitamineum*, which provides a molecular basis for the pathogenic mechanism of *S. scitamineum*.

## Figures and Tables

**Figure 1 jof-08-00470-f001:**
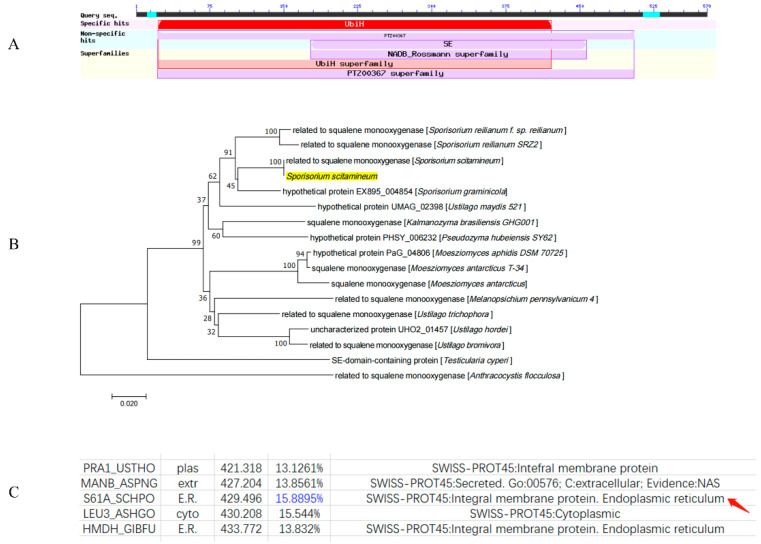
Domain and phylogenetic analysis of the *SsCI80130* gene encoding squalene monooxygenase in *S. scitamineum*. (**A**) The purple “SE” part in the figure showed that predicted protein encoded by the *SsCI80130* gene was squalene monooxygenase. (**B**) Phylogenetic tree analysis of the protein encoded by the *SsCI80130* gene; Arabic numerals at the node edge of the tree indicate the degree of credibility of the phylogenetic tree (using 1000 bootstrap replicates). The protein encoded by the *SsCI80130* gene is highlighted in yellow. (**C**) The subcellular localization prediction of the protein encoded by the *SsCI80130* gene. The result of predicted localization is marked with red arrow. The maximum percentage of matching results is marked in blue.

**Figure 2 jof-08-00470-f002:**
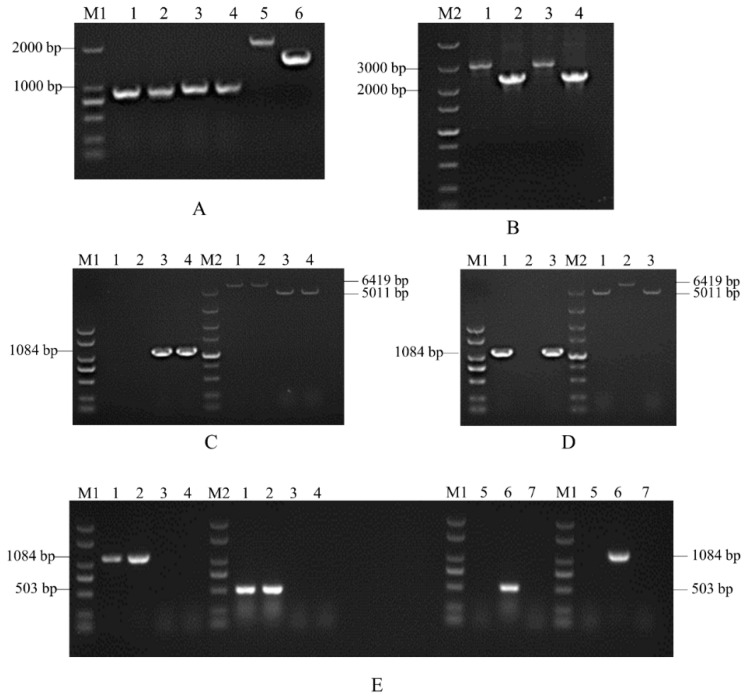
Construction and validation of the *SsCI80130* gene knockout mutants and complementary mutants. Lane M1 is the 2000 marker and lane M2 is the 5000 marker (**A**) PCR amplification. Lanes 1, 2, 3, and 4 represent the left and right flanking fragment under the wild-type (*Ss16*^+^ and *Ss16*^−^) background, respectively; lanes 5 and 6 represent the upstream and downstream fragment of the *Hpt* gene, respectively. (**B**) Fusion-PCR. Lanes 1 and 3 represent the fusion fragments of the left flanking fragment and the upstream *Hpt* fragment under the wild-type background, respectively, while lanes 2 and 4 represent the fusion fragments of the right flanking fragment and downstream *Hpt* fragment, respectively. (**C**) Validation of the knockout mutant *ΔSs80130*^+^ under the *Ss16*^+^ background. In (**C**), lanes 1–3 are transformants and lane 4 represents the wild-type *Ss16^+^*. Lane 1 and 2 are both the knockout mutant *ΔSs80130*^+^. (**D**) Validation of the knockout mutant *ΔSs80130*^−^ under the *Ss16*^−^ background. In (**D**), lane 2 and lane 3 are transformants and lane 1 represents the wild-type *Ss16*^−^. Lane 2 was the knockout mutant *ΔSs80130*^−^. (**E**) Validation of the complementary mutants (*COM80130^+^* and *COM80130*^−^). In the left half, lane 4 is the knockout mutant *ΔSs80130*^+^ and lane 1–3 represent transformants under the *ΔSs80130*^+^ background. In the right half, lane 7 is the knockout mutant *ΔSs80130*^−^ and lane 5–6 represent transformants under the *ΔSs80130*^−^ background. (**E**) denotes that lane 1 and lane 2 are both *COM80130*^+^, and lane 6 is *COM80130*^−^.

**Figure 3 jof-08-00470-f003:**
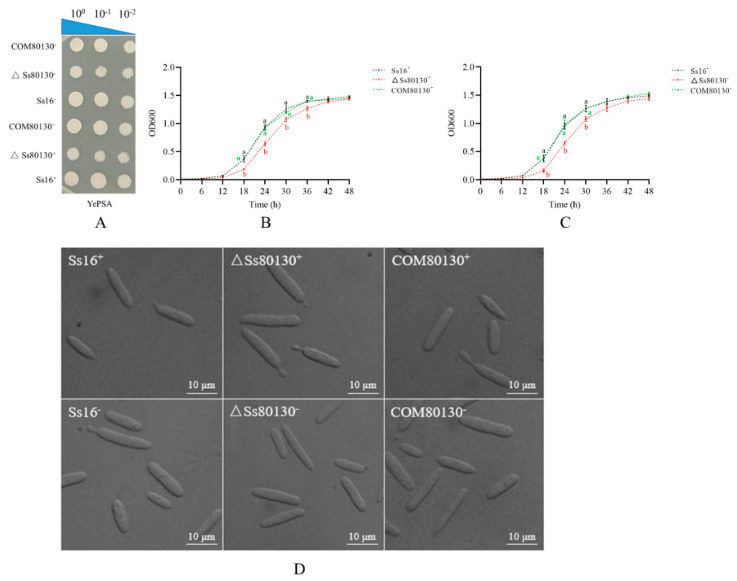
Effects of the *SsCI80130* gene on the haploid phenotype of *S. scitamineum*. (**A**) Colony morphology. Haploid sporidial colonies of the wild type (*Ss16^+^* and *Ss16*^−^), knockout mutants (*ΔSs80130*^+^ and *ΔSs80130*^−^), and complementary mutants (*COM80130^+^* and *COM80130*^−^) were spotted on YePSA plates and incubated at 28 °C for 48 h. (**B**,**C**) Haploid sporidial growth curve. Sporidia of the wild type, knockout mutants, and complementary mutants were inoculated to YePS liquid medium and cultured for 48 h. Each sample had three replicates. Bars indicate the standard error, and different lowercase letters represent a difference at the 0.05 level. (**D**) Microscopic images of sporidia of the wild type, knockout mutants, and complementary mutants (observed under a 100× oil microscope after culturing in YePS liquid medium at 28 °C for 48 h).

**Figure 4 jof-08-00470-f004:**
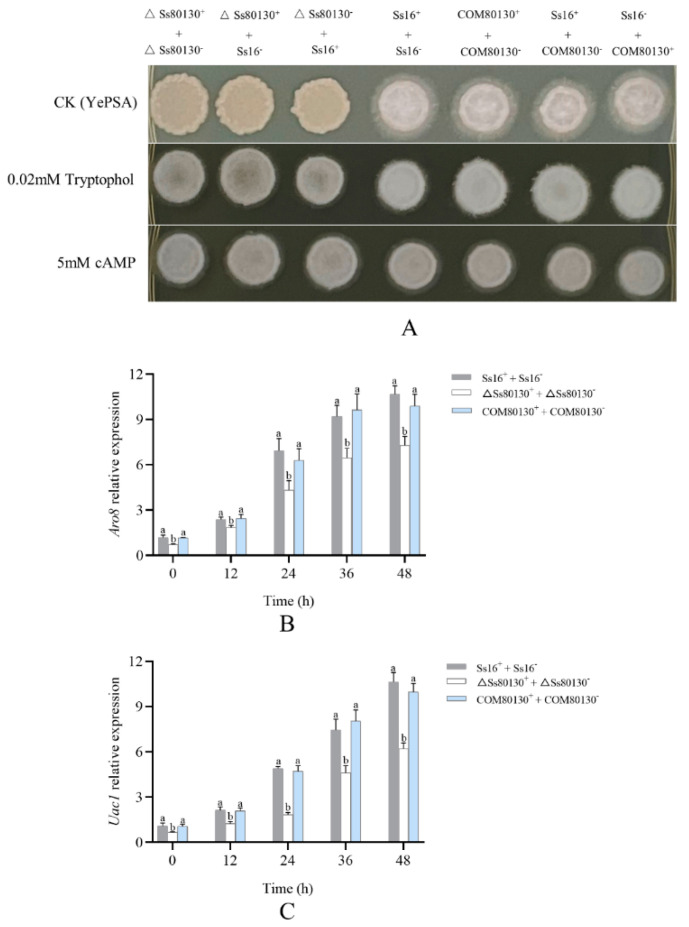
Effects of the *SsCI80130* gene on the sexual mating of *S. scitamineum.* (**A**) Sexual mating ability was assessed on YePSA plates supplemented with cAMP (5 mM) or tryptophol (0.02 mM). Photographs were taken 42 h after inoculation and the control was untreated YePSA plates. (**B**) The expression level of the *Aro8* gene. (**C**) The expression level of the *Uac1* gene. Bars indicate the standard error and different lowercase letters represent a difference at the 0.05 level.

**Figure 5 jof-08-00470-f005:**
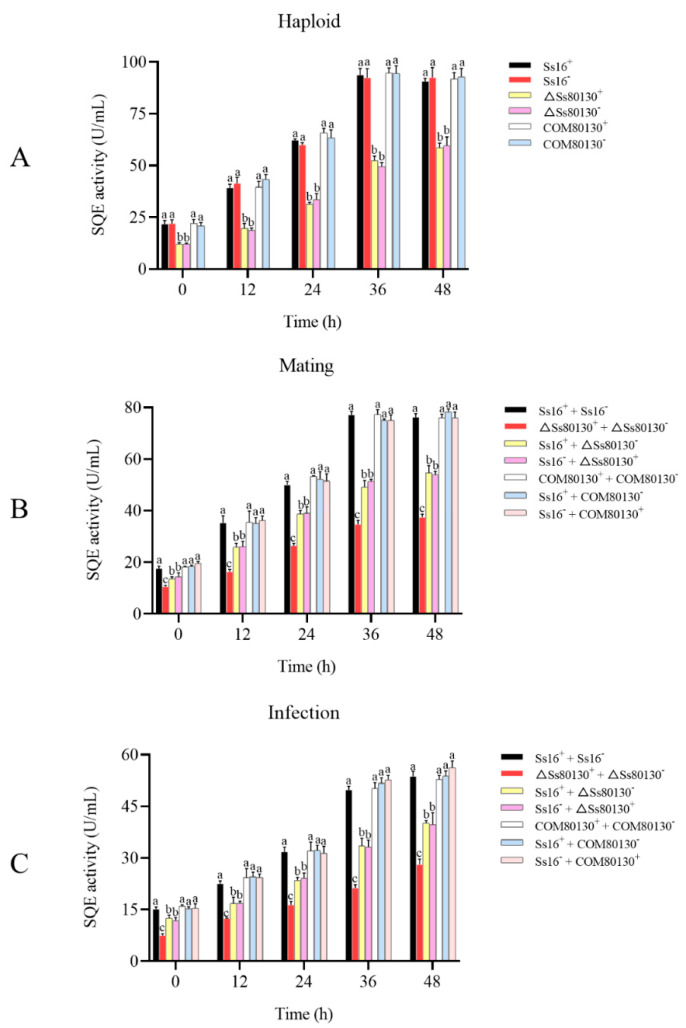
Effects of the *SsCI80130* gene on SQE activity. (**A**) SQE activity during haploid sporidial culture. (**B**) SQE activity during sexual mating culture. (**C**) SQE activity during sugarcane bud infection. Bars indicate the standard error, and different lowercase letters represent a difference at the 0.05 level.

**Figure 6 jof-08-00470-f006:**
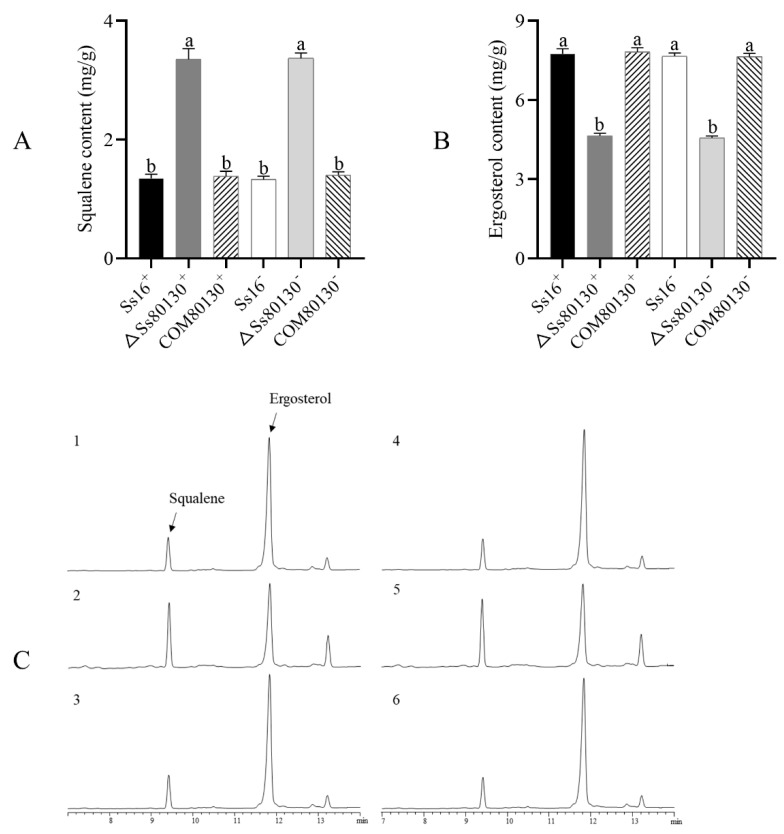
Effects of the *SsCI80130* gene on the squalene and ergosterol content in *S. scitamineum*. (**A**) Squalene content; (**B**) ergosterol content. Bars indicate the standard error, and different lowercase letters represent a difference at the 0.05 level. (C) Gas chromatograms: 1–3 are *Ss16^+^*, *ΔSs80130^+^*, and *COM80130^+^*; 4–6 are *Ss16**^−^*, *ΔSs80130**^−^*, and *COM80130**^−^*.

**Figure 7 jof-08-00470-f007:**
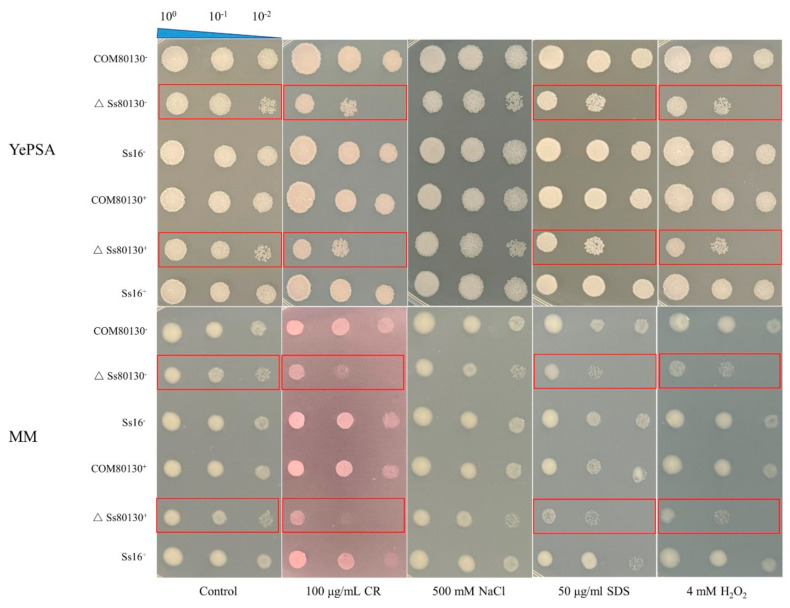
Effects of the *SsCI80130* gene on abiotic stress in *S. scitamineum*. Serially diluted sporidia of the wild type (*Ss16^+^* and *Ss16*^−^), knockout mutants (*ΔSs16^+^* and *ΔSs16*^−^), and complementary mutants (*COM80130^+^* and *COM80130*^−^) were spotted on YePSA plates and MM plates supplemented with Congo red (CR) (100 μg/mL), NaCl (500 mM), SDS (50 μg/mL), or H_2_O_2_ (4 mM). Samples were cultured at 28 °C for 48 h before photographing. Changes in growth of knockout mutants are marked with red frames.

**Figure 8 jof-08-00470-f008:**
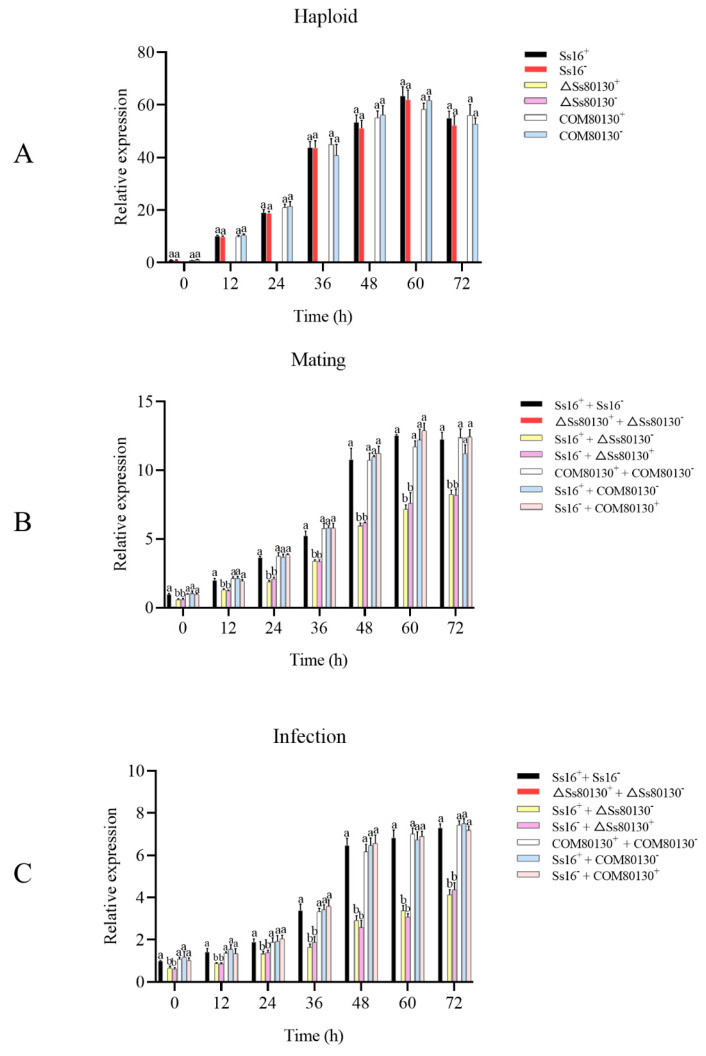
Expression level of the *SsCI80130* gene during various stages. (**A**) Gene expression level during haploid sporidial growth. (**B**) Gene expression level during sexual mating culture. (**C**) Gene expression level during sugarcane bud infection. Bars indicate the standard error, and different lowercase letters represent a difference at the 0.05 level.

**Figure 9 jof-08-00470-f009:**
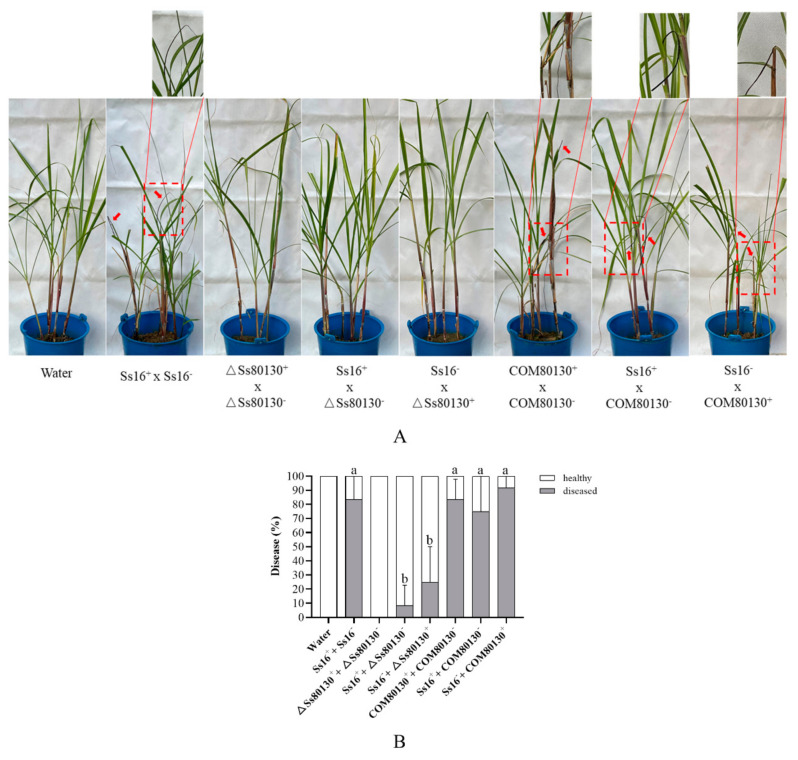
Effects of the *SsCI80130* gene in the pathogenicity of *S. scitamineum*. (**A**) Symptoms of sugarcane smut. The black whips are marked with a red arrow, and the enlarged view of black whips are marked with a red dashed box. (**B**) Final incidence counted after four months of cane plant growth. Bars indicate the standard error. Different lowercase letters represent a difference at the 0.05 level.

**Table 1 jof-08-00470-t001:** Primers used in this study.

Name	Primer Sequences (5′-3′)	Description
*SsCI80130*-LB-F	CTCAAAGCGGCCATCCTTG	
*SsCI80130*-LB-R	GTCGTGACTGGGAAAACCCTGAGACTTGGCGG GATTGTCAC	
*SsCI80130*-RB-F	GGTCATAGCTGTTTCCTGTGTGAGTGCGAGATT GGTGTCGGTAG	Deletion
*SsCI80130*-RB-R	TGAAGATGAGCTCGTTGGGC	construction
Hpt-LB-F	CAGGGTTTTCCCAGTCACGAC	
Hpt-LB-R	GGTCAAGACCAATGCGGAGC	
Hpt-RB-F	GCAAGACCTGCCTGAAACCG	
Hpt-RB-R	TCACACAGGAAACAGCTATGACC	
*SsCI80130*-IN-F *SsCI80130*-IN-R *SsCI80130*-OU-F *SsCI80130*-OU-R	TGGACTTCATCTCTGTGAACAC CGAGATCTACTCAAGGACGAAC TCTCAACAAGGGGGGCTACT GTTTTCGCGTGTGCTGATC	PCR verification
80130COM-F 80130COM-R COM-HPT-LB-F Zeocin-R Situ-F COM-HPT-RB-R	ATCCAAGCTCAAGCTAAGCTTATACACTCGCCAC ATCGGTGC CAGCAAGATCTAATCAAGCTTGGGACAGAGCATA GGACAGCC GCGCGCGTAATACGACTCAC GAAGTGCACGCAGTTGCCG CTCCGTGTTGATGCTGGGAC CGAGCATTCACTAGGCAACCA	Complementation construction
Zeocin-IN-F Zeocin-IN-R	CTGTGATCAGCAGCCAAT GTCAACTTGGCCATGGTG	PCR verification
*SsCI80130*-qF	GATCTCATGATGCCGGCAGA	
*SsCI80130*-qR *Aro8*-qF *Aro8*-qR *Uac1*-qF *Uac1*-qR	TGCTTCGTAGGGTGTGACAC CCTGGTGTTGCGTTCATTCC CAAGCTCGGGCATCGTCTTA CTGACGGAGATGTAGCCAAAG AACGAGACAAGGAGGGAGTA	qRT-PCR
*Actin*-qF	ACAGGACGGCCTGGATAG	
*Actin*-qR	TCACCAACTGGGACGACA	

## Data Availability

Not applicable.

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
