# Peer review of "Squalene Monooxygenase Gene SsCI80130 Regulates Sporisorium scitamineum Mating/Filamentation and Pathogenicity"

_jof, 2022, doi:10.3390/jof8050470_

Round 1

Reviewer 1 Report

The authors researched on sugarcane smut pathogen Sporisorium scitamineum SsCI80130 gene that affects sexual mating and found that it regulates pathogenicity by  the ergosterol synthesis and the synthesis of small-molecular signal substance cAMP or tryptophol required for sexual mating. Further, they performed phenotypic analysis with knockout and complementary mutants, identified pathogenicity  of the biological function of SsCI80130 gene in S. scitamineum.

Author Response

Thank you for your kind comments on our research and manuscript. We have spell-checked the full text of the manuscript.

Reviewer 2 Report

Abstract is not well clear defined the work

There is 54% similarity in the text with others. 

There is no talk about design of experiment, replicants ??????????

Introduction is not to the extend to to cover the related subject.

Material and methods not defined the extent of the work properly.

Results are too lengthy and lots of Figures, of which few must pushed to supplementary.

Discussion having long paragraphs, not discussed properly with other related works.

Conclusion not giving a clear cut summery and the extent of the novelty.

Round 2

Reviewer 2 Report

Its ok by now

Author Response

Thank you for your kind comments on our revised manuscript. We have refined the English expression of the manuscript. Marked in red, lines 53-64, line 235-277, lines 316-429, and lines 465-508 in the revised manuscript (PDF version).